# PERFORMANCE PREDICTION VIA UNSUPERVISED DOMAIN ADAPTATION FOR ARCHITECTURE SEARCH

## ABSTRACT

Performance predictors can directly predict the performance value of given neural architectures without training, thus broadly being studied to alleviate the prohibitive cost of Neural Architecture Search (NAS). However, existing performance predictors still require training a large number of architectures from scratch to get their performance labels as the training dataset, which is still computationally expensive. To solve this issue, we develop an performance predictor by applying the unsupervised domain adaptation technique called USPP, which can avoid costly dataset construction by using existing fully-trained architectures. Specifically, a progressive domain-invariant feature extraction method is proposed to assist in extracting domain-invariant features due to the great transferability challenge caused by the rich domain-specific features. Furthermore, a learnable representation (denoted as operation embedding) is designed to replace the fixed encoding of the operations to transfer more knowledge about operations to the target search space. In experiments, we train the predictor by the labeled architectures in NAS-Bench-101 and predict the architectures in the DARTS search space. Compared with other state-of-the-art NAS methods, the proposed USPP only costs 0.02 GPU days but finds the architecture with 97.86% on CIFAR-10 and 76.50% top-1 accuracy on ImageNet.

## 1 INTRODUCTION

Neural Architecture Search (NAS) (Elsken et al., 2019) aims to automatically design high-performance neural architectures and has been a popular research field of machine learning. In recent years, the architectures searched by NAS have outperformed manually-designed architectures in many fields (Howard et al., 2019; Real et al., 2019). However, NAS generally requires massive computation resources to estimate the performance of architectures obtained during the search process (Real et al., 2019; Zoph et al., 2018). In practice, this is unaffordable for most researchers interested. As a result, how to speed up the estimation of neural architectures has become a hot topic among the NAS community.

Performance predictor (Wen et al., 2020) is a popular accelerated method for NAS. It can directly predict the performance of neural architectures without training, thus greatly accelerating the NAS process. A large number of related works are carried out because of its superiority in reducing the costs of NAS. For example, E2EPP (Sun et al., 2019) adopted a random forest (Breiman, 2001) as the regression model to effectively find promising architectures. ReNAS (Xu et al., 2021) used a simple LeNet-5 network (LeCun et al., 1998) as the regression model, and creatively employed a ranking-based loss function to train the predictor, thus improving the prediction ability of the performance predictor. Although existing performance predictors gain huge success in improving the efficiency of NAS, sufficient architectures need to be sampled from the target search space and be fully trained to obtain their performance value as the label (Wen et al., 2020). The performance predictor is trained by these labeled architectures, and then is used to predict the performance of architectures. In order to ensure the prediction accuracy of the performance predictor, it is usually necessary to train at least hundreds of architectures as the dataset, which is a huge cost.

In recent years, many benchmark datasets such as NAS-Bench-101 (Ying et al., 2019), NAS-Bench-201 (Dong & Yang, 2020), NAS-Bench-NLP (Klyuchnikov et al., 2020) are released for promoting the research on NAS. There are a large number of architecture pairs (i.e., the architecture and its

performance) in these datasets. As a result, we are motivated to utilize the rich architecture knowledge in these datasets to predict the architectures in the target search space (i.e., the search space in which the architecture needs to be predicted.). In this way, we can avoid training a large number of architectures in the target search space, thereby alleviating the expensive cost of building the dataset for performance predictors. However, the search space designed in the benchmark datasets is very different from the real-world search spaces. The performance predictor trained on existing labeled architectures cannot be applied to the target search space.

In this paper, we proposed an UnSupervised domain adaptation-based Performance Predictor (USPP) with the usage of the domain adaptation technique. Different from the traditional performance predictors that need the training data and the predicted data in the same search space, USPP can leverage the labeled architectures in existing benchmark datasets (e.g., NAS-Bench-101 (Ying et al., 2019)) to build a powerful performance predictor for the target search space (e.g., the DARTS search space (Liu et al., 2018b)). As a result, USPP can avoid expensive data collection for the target search space. Specifically, the contributions can be summarized as follows:

- A progressive domain-invariant feature extraction method is proposed to reduce the transfer difficulty caused by the huge difference between source and target search spaces. The progressive method explicitly models the domain-specific features and gradually separates them from the domain-invariant features, thus assisting in the alignment of the source and target search spaces.

- A learnable representation for the operations in architectures, i.e., operation embedding, is designed to transfer more knowledge about operations to the target search space. Compared to the widely used fixed encoding method, the operation embedding can effectively capture the inner meaning and structural role of each operation in the source search space and applied them in the target search space to reduce transfer difficulty.

- USPP only costs $0.02$ GPU days to search for the architectures in the DARTS search space because there is no need to annotate the architectures in the target search space. Furthermore, the searched architecture by USPP achieves $97.86\%$ classification accuracy on CIFAR-10 and $76.50\%$ classification accuracy on ImageNet and outperforms all the state-of-the-art methods compared.

## 2 RELATED WORK

### 2.1 NAS AND PERFORMANCE PREDICTORS

NAS can automatically design high-performance neural architecture and consists of search space, search strategy, and performance estimation strategy (Elsken et al., 2019). Specifically, the search space defines the collections of the candidate architectures. The search strategy corresponds to the employed optimization algorithms for the search, which can be mainly classified into evolutionary algorithms (Bäck et al., 1997), reinforcement learning (Kaelbling et al., 1996) and gradient descent (Liu et al., 2018b). The performance estimation strategy defines how to evaluate the architectures to obtain their performance. During the search, the NAS algorithms use the search strategy to search architecture in the predefined search space and obtain the performance value of the searched architectures by the performance estimation strategy.

No matter what search strategy is used, a lot of neural architectures need to be estimated. Because of the heavy cost of the traditional GPU-based estimation method, many accelerated methods are proposed such as early stopping policy (Sun et al., 2018), proxy dataset (Sapra & Pimentel, 2020), weight-sharing method (Bender et al., 2018). However, the first two methods may lead to poor generalization and low-fidelity approximation of performance value, and the weight-sharing method may be unreliable in predicting the relative ranking among architectures (Li et al.; Yang et al.), which disobeys the goal of finding the architecture with the highest ranking. Performance predictor is free from the aforementioned shortcomings and has received great attention in recent years. However, existing predictors have the constraint that the source search space and the target search space must be the same. The proposed USPP breaks the limitation and creatively uses the existing labeled architectures in the source search space to predict the architectures in the target search space, thus removing the reliance on potentially costly labels in the target search space.

## 2.2 DOMAIN ADAPTATION

In many machine learning methods, there is a major assumption that the training data and testing data are from the same distribution (Wilson & Cook, 2020). When it is not held, the network trained on the source domain will face a performance decline when testing on the target domain (Patel et al., 2015). Transfer learning (Weiss et al., 2016) can solve the problem by learning knowledge from the source domain and applying it to the target domain. Domain Adaptation (DA) (Wilson & Cook, 2020) is a subcategory of transfer learning, and its goal is to train a network that performs well on a different but related target domain by the labeled data in the source domain. The scenario in this paper is a problem of Unsupervised Domain Adaptation (UDA), where the labeled data are only available in the source domain.

The mainstream method to address the UDA is aligning source and target domains by learning the domain-invariant feature representation. Specifically, the features are said to be domain-invariant if features extracted from data in both source and target domains follow the same distribution. If a network performs well in the source domain using domain-invariant feature representation, the network can be generalized well to the target domain. Generally, the domain-invariant methods can be classified into two categories. The first category explicitly reduces the domain discrepancy to obtain the domain-invariant by some distribution discrepancy metrics such as Maximum Mean Discrepancy (MMD) (Gretton et al., 2006; 2012), correlation alignment (Sun et al., 2016), and contrastive domain discrepancy (Kang et al., 2019). Motivated by the Generative Adversarial Network (GAN) (Goodfellow et al., 2014), another category learns domain-invariant representation by adversarial training (Ganin et al., 2016), and this is also what our method belongs to. The methods generally train a domain discriminator to distinguish the source domain from the target domain, and train a feature extractor to fool the discriminator to extract domain-invariant representations. Adversarial domain adaptation achieves great success in many fields, and a lot of works are proposed. For example, symnets (Zhang et al., 2019) designed symmetric classifiers to play the role of domain discriminator. GVB (Cui et al., 2020b) constructed bridge layers on both the generator and discriminator to reduce the overall transfer difficulty. However, most existing adversarial methods are performed on the classification task of Computer Vision (CV) or Neural Language Processing (NLP), and cannot be directly exploited in performance predictor which is a regression problem and deals with a completely different data type (i.e., graph data). In this paper, we apply the adversarial methods in performance predictors and propose a progressive domain-invariant feature extraction method to reduce the transferring difficulty. As far as we know, this is the first work to alleviate the high costs of performance predictors by the adversarial domain adaptation techniques.

## 3 APPROACH

### 3.1 FORMULATION

In the scenarios of our paper, the architecture dataset in the source domain is denoted as $\mathcal{D}_{\mathcal{S}} = \left\{ \left( \mathbf{x}_i^S, \mathbf{y}_i^S \right) \right\}_{i=1}^{N_s}$ with $N_S$ labeled architectures, where $\mathbf{x}_i^S$ and $\mathbf{y}_i^S$ denote the $i^{th}$ architecture and its performance value, respectively. The dataset in the target domain is denoted as $\mathcal{D}_{\mathcal{T}} = \left\{ \mathbf{x}_i^T \right\}_{i=1}^{N_T}$ with $N_T$ unlabeled architectures. USPP aims to train a performance predictor on $\mathcal{D}_{\mathcal{S}}$ and $\mathcal{D}_{\mathcal{T}}$, and achieve promising results in the performance prediction of the architectures in target domain.

The overall framework of the proposed USPP is shown in Fig 1. Specifically, to build the predictor, the neural architectures first should be represented into a form that can be fed into the predictor. Generally speaking, most neural architectures can be regarded as the Directed Acyclic Graph (DAG). The node in the DAG corresponds to one specific operation (e.g., convolution 3×3), and the edge stands for the connection between operations. Consequently, each architecture can be represented by the adjacency matrix $\mathbf{A}$ and the operation list $L_{op}$, where $\mathbf{A}$ stands for the topology connections between nodes and $L_{op}$ represents the operations of nodes. Because the most commonly used encoding method for $L_{op}$ (i.e., one-hot method) ignores the intrinsic relationship between different operations, we adopt a learnable operation embedding $\mathcal{E}$ to map $L_{op}$ to a continuous space $\mathbb{R}$ to avoid the drawback, and the operation embedding will be elaborated in Sec. 3.4. Then, we use a Graph Convolution Network (GCN) as **the first component** of USPP to map every architecture into the feature space $\mathcal{Z}$. In specific, GCN is a powerful technique to learn useful features from graph data which is a kind of non-Euclidean data, and has achieved success in processing neural architecture

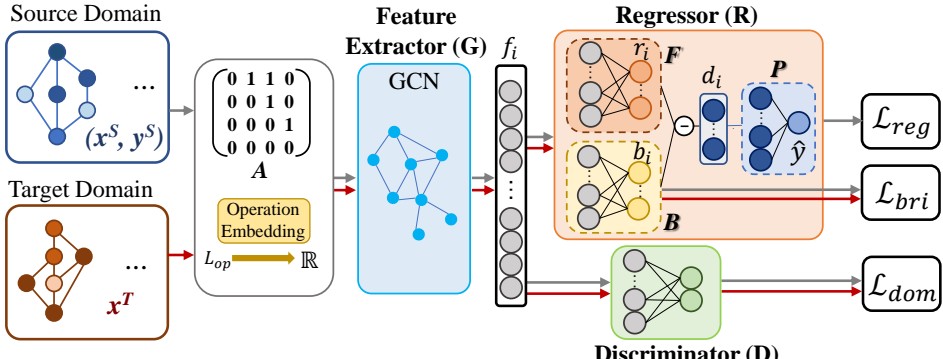

Figure 1: The overall framework of USPP.

data. After learning the latent representation of the architectures by $G$, regressor $R$, as **the second component** of USPP, is exploited to predict the performance of the architectures.

To ensure the prediction accuracy in target domain, the prediction results in source domain needs to be guaranteed first. It can be achieved by minimizing the regression loss $\mathcal{L}_{reg}$ on $G$ and $R$ when training. The regression loss can be calculated as:

$$\mathcal{L}_{reg} = \frac{1}{N_S} \sum_{i=1}^{N_S} \mathcal{L}\big(\mathbf{y}_i^S, R\left(G\left(\mathbf{x}_i^S\right)\right)\big) \tag{1}$$

where $\mathcal{L}$ is the $L_2$ loss fuction in USPP.

## 3.2 ADVERSARIAL TRAINING

As mentioned above, we are motivated to use the existing labeled architectures in the benchmark datasets to build a performance predictor for the target search space. However, the gap between the architectures in different search spaces is so large that the performance predictor trained with the labeled architectures in the source domain cannot be directly applied to predict the architectures in the target domain. Accordingly, we leverage the adversarial training (Wilson & Cook, 2020) to learn domain-invariant representations, thus reducing the discrepancy between source and target domains. In this way, the performance predictor can generalize well to the target domain.

To perform adversarial training, a discriminator $D$ is designed as **the third component** of USPP, and it is a fully connected layer. Similar to the regressor $R$, $D$ is attached after the feature extractor $G$. Specifically, the discriminator is essentially a domain classifier to differentiate between the features from the source domain and the target domain, and only works during the training process of USPP. During the training, the discriminator $D$ is optimized to correctly classify the domains, while the feature extractor $G$ is optimized to fool the discriminator so that the discriminator cannot distinguish the source domain from the target domain. Through the adversarial training, the feature extractor is encouraged to extract the common features shared by domains (i.e., domain-invariant features).

To adversarially train the discriminator and the feature extractor, the domain classification loss $\mathcal{L}_{cls}$ is designed, which can be formulated as:

$$\mathcal{L}_{cls} = -\frac{1}{N_S} \sum_{i=1}^{N_S} \log\left(D\left(G\left(\mathbf{x}_i^S\right)\right)\right) - \frac{1}{N_T} \sum_{j=1}^{N_T} \log\left(1 - D\left(G\left(\mathbf{x}_j^T\right)\right)\right) \tag{2}$$

To make $D$ distinguish between the source domain and target domain, we train $D$ to minimize the domain classification loss $\mathcal{L}_{cls}$. At the same time, we train $G$ to maximize $\mathcal{L}_{cls}$, thus making the feature distributions from source and target domains as similar as possible to fool $D$.

## 3.3 PROGRESSIVE DOMAIN-INVARIANT FEATURE EXTRACTION

Although the adversarial training can align the source and target domains to a certain extent, it is difficult to reduce the divergence between source and target domains to zero in practice due to

the existence of rich domain-specific features in respective domains. Specifically, domain-specific features refer to the features unique to each domain and greatly hinder the alignment of the data distributions from source and target domains. To further reduce the negative influence of domain-specific features, we propose a simple but effective progressive domain-invariant feature extraction method.

The method reduces the influence of domain-specific features mainly by explicitly modeling them. In fact, there are some similar works to mitigate domain-specific features in this way. For example, DSN (Bousmalis et al., 2016) separately modeled the domain-specific features of each domain and the domain-invariant feature shared by source and target domains and reconstructed the input data by the extracted domain-specific and domain-invariant features to add generalizability. However, the modeled domain-invariant features also participated in the reconstruction, leading to them with a lot of domain-specific properties. HDA (Cui et al., 2020a) proposed a heuristic framework, and leveraged the modeled domain-specific features as heuristics to gradually gain domain-invariant features. However, the multiple sub-networks in the heuristic network made it difficult to optimize. The proposed progressive feature extraction method can effectively model the domain-specific features and gradually separate them from the features for the architecture data processed in this paper.

To alleviate the domain-specific features more easily, we hypothesize that the domain-specific features are easier to be captured compared with the domain-invariant features according to HDA. This is because the architectures in the same domain generally share common domain-specific characteristics. For example, the cell in the NAS-Bench-101 only has one input while that in the DARTS search space has two inputs. Furthermore, the operations in NAS-Bench-101 are completely different from those in the DARTS search space. Compared with the domain-invariant features, these domain-specific features are easier to be extracted. The proposed method is embodied in the design of the regressor $R$. The regressor is used to map the extracted feature from $G$ to the predicted label, and consists of fundament layer $F$, bridge layer $B$, and prediction layer $P$. Specifically, fundament layer $F$ is used to model the current feature representation $r_i$, and bridge layer $B$ is utilized to explicitly model the discrepancy from the current feature to the ideal domain-invariant feature, i.e., the domain-specific part $b_i$ in the current feature. By subtracting the results of bridge layer $b_i$ from the results of $r_i$, the domain-invariant features $d_i$ can be obtained. This is because $b_i$ is easier to be obtained than $d_i$ and can give guide to the constructing of $d_i$. Finally, the obtained $d_i$ is fed to the prediction layer $P$ to get the final prediction label $\hat{y}$. The overall process can be formulated as:

$$R(f_i) = P(F(f_i) - B(f_i)) \tag{3}$$

where $f_i$ denotes the extracted feature from $G$. During training, we seek to reduce the influence of the modeled domain-specific feature $b_i$ to help the extraction of domain-invariant representation. To achieve this goal, we design an extra loss $\mathcal{L}_{bri}$ as following:

$$\mathcal{L}_{bri} = \frac{1}{(N_S + N_T)} \sum_{i=1}^{N_s+N_t} \sum_{j=1}^{L} |b_{i,j}| \tag{4}$$

where $L$ is the size of $b_i$. During the training of USPP, we aim to gradually minimize $\mathcal{L}_{bri}$, thus mitigating domain-specific representation. Therefore, at the beginning of training, the designed bridge layer assists in the progressive extraction of the domain-invariant feature. In the later stage of training, $b_i$ is close to zero, which will not have too much impact on training.

In conclusion, USPP has three objectives in the training phase: 1. Minimizing $\mathcal{L}_{reg}$ on $G$ and $R$ to get high prediction accuracy in the source domain. 2. Minimizing $\mathcal{L}_{cls}$ on $D$ and maximizing $\mathcal{L}_{cls}$ on $G$ to perform adversarial training. 3. Minimizing $\mathcal{L}_{bri}$ on $R$ and $G$ to assist in the extraction of the domain-invariant features. Let $\mathcal{L}_{dom} = -\mathcal{L}_{cls}$, the overall objectives during training is as following:

$$\begin{aligned} \min_{G,R} \quad & \mathcal{L}_{reg} + \gamma\mathcal{L}_{dom} + \mu\mathcal{L}_{bri}; \\ \max_{D} \quad & \mathcal{L}_{dom} \end{aligned} \tag{5}$$

where $\gamma$ and $\mu$ are the hyper-parameters.

### 3.4 OPERATION EMBEDDING

As is mentioned above, neural architectures can be represented by the adjacency matrix $\boldsymbol{A}$ and the operation list $L_{op}$, where $\boldsymbol{A}$ represents the connection relationship between different nodes, and

$L_{op}$ represents the operation type of every node. Generally, the one-hot method is used to encode the operation list $L_{op}$ to a fixed matrix $\mathcal{M}^{len(L_{op}) \times K}$ (Liu et al., 2021). Then, the fixed matrix $\mathcal{M}^{len(L_{op}) \times K}$ and $\boldsymbol{A}$ are fed into GCN to obtain the feature representation of a given architecture. However, there is a large limitation that the process ignores the computational relationship between different operations. This is because every operation is thought to be independent in the one-hot method, but it is obviously not in line with reality. For example, the role of $1 \times 1$ convolution may be more similar to that of $3 \times 3$ convolution compared with that of the max pooling operation. According to this motivation, the trainable operation embedding is proposed instead of the one-hot method.

Specifically, we use operation embedding $\mathcal{E}$ to map each operation in $L_{op}$ to a continuous space $\mathbb{R}^{len(L_{op}) \times K}$. $K$ is the dimension of the mapped vector and can be set during the training. Then, the $K$-dimensional vector obtained by $\mathcal{E}$ along with the adjacency matrix is mapped to the feature representation by GCN. During the training process of USPP, the operation embedding is regarded as a part of GCN, and the weight of the operation embedding is optimized with the weights of GCN. Through this, the operation embedding $\mathcal{E}$ can seek the inner relationship between different operations during the training. In specific, the operation embedding is similar to the word embedding in NLP, except that the mapped object words are replaced by the operation in the architecture. Word embedding can encode the semantic and syntactic information of every word, where semantic information is related to the meaning of words, and syntactic information corresponds to the structural roles of words (Li & Yang, 2018). Similarly, the inner meaning and structural information of each operation can be also learned from the source domain and applied in the target domain through the operation embedding. As a result, operation embedding helps USPP transfer more information about the operation from the source domain to the target domain, thus improving the prediction performance in the target domain.

## 3.5 THEORETICAL ANALYSIS

In this section, we analyze the expected target-domain error for the proposed method based on the theories of domain adaptation (Ben-David et al., 2010). Let $\mathcal{S}$ and $\mathcal{T}$ be the source and target distributions for a family of fundament layer $F$ in the hypothesis space $\mathcal{H}$. $\varepsilon_{\mathcal{T}}(F)$ and $\varepsilon_{\mathcal{S}}(F)$ be the expected error of a hypothesis $F \in \mathcal{H}$ on target and source domains. To bound the target error, we use the $\mathcal{H}\Delta\mathcal{H}$-distance to measure the distance between distributions $\mathcal{S}$ and $\mathcal{T}$, which is defined as the following:

$$d_{\mathcal{H}\Delta\mathcal{H}}(\mathcal{S},\mathcal{T}) = 2 \sup_{h_1,h_2 \in \mathcal{H}} | P_{\mathbf{f}\sim\mathcal{S}}[h_1(\mathbf{f}) \neq h_2(\mathbf{f})] - P_{\mathbf{f}\sim\mathcal{T}}[h_1(\mathbf{f}) \neq h_2(\mathbf{f})] | \tag{6}$$

We can get the target error of hypothesis $F$ by the theory in (Ben-David et al., 2010):

$$\varepsilon_{\mathcal{T}}(F) \leq \varepsilon_{\mathcal{S}}(F) + \frac{1}{2}d_{\mathcal{H}\Delta\mathcal{H}}(\mathcal{S},\mathcal{T}) + \lambda \tag{7}$$

where $\lambda = \varepsilon_T(F^*) + \varepsilon_S(F^*)$ is the combined error of ideal hypothesis $F^* = \arg\min_F(\varepsilon_S(F) + \varepsilon_T(F))$ on both domains. By the definition of $\mathcal{H}\Delta\mathcal{H}$-distance, $d_{\mathcal{H}\Delta\mathcal{H}}(\mathcal{D}_S,\mathcal{D}_T) = 2\sup|\varepsilon_S(h,h') - \varepsilon_T(h,h')|$. Hence,

$$\varepsilon_{\mathcal{T}}(F) \leq \varepsilon_{\mathcal{S}}(F) + \sup|\varepsilon_S(F,F^*) - \varepsilon_T(F,F^*)| + \lambda \tag{8}$$

Because the ideal discrepancy $B^*$ from the current feature to the ideal domain-invariant feature may be not fully represented by the practice discrepancy $B$, $B \leq B^*$. Furthermore, $B^* = F - F^*$. Hence, there exists a positive correlation between $B$ and $F$:

$$B = k(F - F^*) \quad k \in (0,1] \tag{9}$$

As mentioned in Subsec. 3.3, $F = B + P$. Furthermore, the goal of $F$ and $P$ is both to better obtain the performance of the given architecture. As a result, the ideal hypothesis $F^* = P^*$. We could obtain the relationship between $F$ and $P$:

$$(1-k)(F - F^*) = (P - P^*) \tag{10}$$

Equation (10) is held on source and target distributions. Hence, $\varepsilon_S(P,P^*) = (1-k)\varepsilon_S(F,F^*)$ and $\varepsilon_T(P,P^*) = (1-k)\varepsilon_T(F,F^*)$. The $P$ and $F$ can both correctly predict the performance

Table 1: Comparison with other methods in DARTS search space on CIFAR-10. The second rows shows the state-of-the-art NAS method, and the third row shows the advanced performance predictor. The fourth row shows the methods that do not require training architecture as USPP does.

| Architecture | Accuracy(%) | #Params.(M) | Cost (GPU day) |
|---|---|---|---|
| NASNet-A (Zoph et al., 2018) | 97.35 | 3.3 | 1,800 |
| AmoebaNet-A (Real et al., 2019) | 96.66±0.06 | 3.2 | 3,150 |
| ENAS (Pham et al., 2018) | 97.11 | 4.6 | 0.5 |
| DARTS (Liu et al., 2018b) | 97.24±0.09 | 3.4 | 4 |
| SNAS (Xie et al., 2018) | 97.02 | **2.9** | 1.5 |
| PC-DARTS (Xu et al., 2019) | 97.43 | 3.6 | 0.1 |
| P-DARTS (Chen et al., 2019) | 97.50 | 3.4 | 0.3 |
| DARTS-PT (Wang et al., 2021b) | 97.54 | 3.9 | 0.29 |
| RANK-NOSH (Wang et al., 2021a) | 97.50 | 3.5 | - |
| NAONet (Luo et al., 2018) | 97.02 | 28.6 | 200 |
| BONAS (Shi et al., 2020) | 97.31 | 3.45 | 2.5 |
| CTNAS (Chen et al., 2021) | 97.41±0.04 | 3.6 | 0.3 |
| TNASP (Lu et al., 2021a) | 97.43±0.04 | 3.6±0.1 | 0.3 |
| NASWOT (Mellor et al., 2021) | 95.73 | 4.96 | - |
| TENAS (Chen et al., 2020) | 97.37±0.064 | 3.8 | 0.05 |
| **USPP(ours)** | **97.86** | 3.3 | **0.02** |

of architectures, the differences between them on the source domain are small. Hence, $\varepsilon_S(P) = \varepsilon_S(F)$. Combined with Equation (10), we can obtain Equation (11):

$$
\begin{aligned}
\varepsilon_T(P) &\leq \varepsilon_S(P) + \sup |\varepsilon_S(P, P^*) - \varepsilon_T(P, P^*)| + \lambda \\
&\leq \varepsilon_S(F) + (1 - k) \sup |\varepsilon_S(F, F^*) - \varepsilon_T(F, F^*)| + \lambda \\
&\leq \varepsilon_S(F) + \frac{(1-k)}{2} d_{\mathcal{H} \Delta \mathcal{H}}(\mathcal{S}, \mathcal{T}) + \lambda
\end{aligned}
\tag{11}
$$

It can be seen from Equation (11) that the upper bound of the target-domain error for $P$ with the usage of $B$ is lower than that for $F$ without the usage of $B$. Therefore, the proposed method can help to reduce the upper bound of the target-domain error, which also proves the effectiveness of our proposed method from a theoretical point of view.

## 4 EXPERIMENTS

In this part, we choose the NAS-Bench-101 as source domain and the DARTS search space as target domains to train USPP. We first report the results of neural architecture search in DARTS to verify the effectiveness of USPP. Then, we perform extensive ablation studies on various components. Please note that we perform additional experiments (NAS-Bench-101→NAS-Bench-201, NAS-Bench-201→NAS-Bench-101, NAS-Bench-201→DARTS, NDS ResNet→NDS ResNeXt Radosavovic et al. (2019), and NDS ResNeXt→NDS ResNet) in Appendix B. All experiments in the paper are performed on NVIDIA GTX 2080Ti GPU. The experimental settings is presented in Appendix A.

### 4.1 NEURAL ARCHITECTURE SEARCH ON DARTS

**Training process.** The regression loss $\mathcal{L}_{reg}$, the domain classification loss $\mathcal{L}_{cls}$, and the bridge loss $\mathcal{L}_{bri}$ are shown in Fig. 4.1(a). As expected, $\mathcal{L}_{reg}$, $\mathcal{L}_{cls}$, and $\mathcal{L}_{bri}$ reduce repaidly. Kendall's Tau (KTau) (Sen, 1968), which can describe the correlation between the predicted value and the ground-truth values, is used to measure the prediction performance of USPP. The trend of the KTau value of USPP is shown in Fig. 4.1(b). It can be seen that the KTau of USPP goes up very quickly and converges.

**CIFAR-10.** The experimental results on CIFAR-10 are shown in Table 1. It can be seen that the proposed USPP only costs 0.02 GPU days, which is the least among all methods compared. Please

Table 2: Comparison with other methods in DARTS search space on ImageNet. The second rows shows the state-of-the-art NAS method, while the third row shows the advanced performance predictor. The fourth row shows the methods that do not require training architecture as USPP does. The symbol '—' denotes that no results are available in original papers. † represents the results with label smoothing, autoaugment and SE methods.

| Architecture | Top-1(%) | Top-5(%) | #Params.(M) | Cost (GPU day) |
|---|---|---|---|---|
| NASNet-A (Zoph et al., 2018) | 74.0 | 91.6 | 5.3 | 2,000 |
| AmoebaNet-A (Real et al., 2019) | 74.5 | 92.0 | 3.2 | 3,150 |
| PNAS (Liu et al., 2018a) | 74.2 | 91.9 | 5.1 | 225 |
| DARTS (Liu et al., 2018b) | 73.3 | 91.3 | 4.7 | 4 |
| SNAS (Xie et al., 2018) | 72.7 | 90.8 | 4.3 | 1.5 |
| PC-DARTS (Xu et al., 2019) | 74.9 | 92.7 | 3.6 | 0.1 |
| DropNAS (Hong et al., 2021) | 75.5 | 92.6 | 5.2 | 0.6 |
| DARTS-PT (Wang et al., 2021b) | 74.5 | 92.0 | 4.6 | 0.29 |
| NAONet (Luo et al., 2018) | 74.3 | 91.8 | 11.35 | 200 |
| Neural Predictor (Wen et al., 2020) | 74.7 | - | - | - |
| RANK-NOSH (Wang et al., 2021a) | 74.8 | - | 5.3 | 0.3 |
| TENAS (Chen et al., 2020) | 75.5 | 92.5 | 5.4 | 0.17 |
| **USPP(ours)** | 75.9 | 92.9 | 5.0 | **0.02** |
| **USPP(ours)†** | **76.5** | **93.0** | 5.3 | **0.02** |

note that we do not take the costs for building the NAS-Bench-101 into consideration because NAS-Bench-101 is a public dataset and not constructed for our work. Besides, TENAS with the least cost among other methods, takes 2.5 times as long as USPP. This is because USPP takes advantage of the fully-trained architectures in NAS-Bench-101, thus eliminating the cost of building new architectures in the DARTS search space. Furthermore, the searched architecture by USPP gets the highest test accuracy 97.86% compared with all the other state-of-the-art methods. Specifically, the accuracy of the architecture searched is still 0.32% higher than that of the best performance architecture searched by other methods. This shows that our method can effectively transfer knowledge from the source search space to the target search space.

**ImageNet.** The comparison results on ImageNet are shown in Table 2. It can be seen from the results that the search cost of USPP is the lowest, and the PC-DARTS method with the least cost of the other methods is five times of the USPP. Besides, the searched architecture by USPP trained without any tricks can obtain the highest top-1 and top-5 accuracy, and the training result with positive technique largely surpasses existing methods in top-1 and top-5 accuracy. As a result, USPP can transfer useful information from NAS-Bench-101 and learn the inner correspondence between architectures and performance in DARTS.

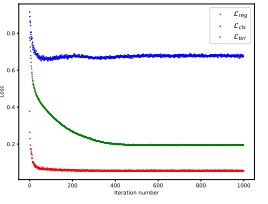

(a) The losses of USPP.

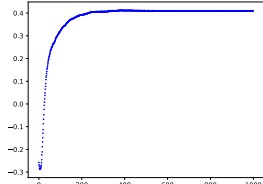

(b) The KTau value of USPP.

Figure 2: The training process on NAS-Bench-101→DARTS.

Table 3: The ablation study on the progressive domain-invariant feature extraction.

| Bridge Layer | Kendall's Tau |
|---|---|
| × | 0.2679 |
| √ | **0.4113** |

Table 4: The ablation study on the operation embedding.

| Encode Method of $L_{op}$ | Kendall's Tau |
|---|---|
| one-hot | 0.3104 |
| operation embedding | **0.4113** |

Table 5: The ablation study on the domain adaptation method.

| Domain adaptation method | Kendall's Tau |
|---|---|
| MMD (Gretton et al., 2006) | 0.3162 |
| DANN (Ganin & Lempitsky, 2015) | 0.2424 |
| GVB (Cui et al., 2020b) | 0.2699 |
| HDA (Cui et al., 2020a) | 0.3173 |
| USPP(ours) | **0.4113** |

## 4.2 ABLATION STUDY

In this part, we conduct ablation experiments on the progressive domain-invariant feature extraction method, operation embedding, and domain adaptation methods. We randomly sample 100 architectures from NAS-Bench-301 Siems et al. (2020) to evaluate the proposed USPP.

**The progressive domain-invariant feature extraction.** To validate the effectiveness of the proposed progressive method, we have performed ablation experiments on the core components of the progressive domain-invariant feature extraction method, i.e., the bridge layer. It can be seen from Table 3 that the performance predictor with the bridge layer gets a higher Kendall's Tau value and 0.1434 more than the result without the bridge. The result indicates that the progressive domain-invariant feature extraction plays a positive role in improving the performance of USPP.

**Operation embedding.** The results of the ablation study on the operation embedding are reported in Table 4. It can be seen from the result with operation embedding is more 0.1009 than that without operation embedding, which illustrates that operation embedding improves the performance of the predictor. Furthermore, the length of operation embedding is set as 3 in the experiment because of the small number of the operations, and the 3D visual result is shown in Fig. 3. Surprisingly, convolution $1 \times 1$ and output are the most similar pair in all pairs of operations. Furthermore, just as we suspected, the distance between convolution operations is smaller than the distance between convolution and pooling operations.

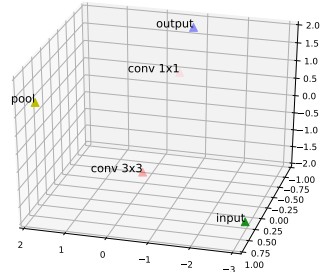

Figure 3: The visual result of operation embedding.

**Different domain adaptation methods.** We adapt the distribution discrepancy metric-based method MMD (Long et al., 2015) and the adversarial training-based methods DANN (Ganin & Lempitsky, 2015), GVB (Cui et al., 2020b), and HDA (Cui et al., 2020a). The hyperparameters of these methods are optimized, and the best results of them are reported in Table 7. It is obvious that our method outperforms all the peer competitors.

## 5 CONCLUSION

In this paper, we propose an performance predictor with the unsupervised domain adaptation technique to reducing the high cost of annotating architectures. Specifically, we design a progressive domain-invariant feature extraction to reduce the difficulty of alignment by explicitly modeling the domain-specific and separating them of domain-invariant features. Moreover, we use operation embedding to map the operation list of every architecture to continuous space, and learn the deeper meaning and structural information of each operation during training. The USPP achieves the test accuracy of 97.86% and 76.5% on CIFAR-10 and ImageNet, respectively. Unfortunately, this paper does not consider sequence-based search spaces and we will propose a more general domain adaptation framework in the future.

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

## A    EXPERIMENTAL SETTING

In this section, we report the expermental settings of the experiments in Section 4.

**Data processing.** In order to keep the DARTS search space and NAS-Bench-101 search space uniform, we convert the architecture of the DARTS search space with operations on edges to one with operations on nodes following the convention (Liu et al., 2021). Then, since the types of operations in the two search spaces are very different, we group all operation types into five categories: input, output, convolution $1 \times 1$, convolution $3 \times 3$, convolution $5 \times 5$, and pooling.

**Setting of USPP.** The Gradient Reversal Layer (GRL) is utilized to simplify the adversarial training. Specifically, GRL can flip the gradients from $D$ to $G$ so that we can train $G$ and $R$ to minimize the sum of $\mathcal{L}_{reg}$, $\mathcal{L}_{dom}$ and $\mathcal{L}_{bri}$. We train the USPP with the batch size of $1,000$ for 100 epoch. We employ the SGD with the weight decay of $0.0005$ and the learning rate is initialed with $0.1$. We set $\gamma = 1$ and $\lambda = 1$ in the loss which is verified to perform well in the experiments.

**Search setting.** Followed the convention of performance predictor (Wen et al., 2020), we randomly sample $100,000$ architectures from the DARTS search space, and predict them by the trained USPP. Then, we select the architecture with the highest predicted performance value as the searched architecture.

**Training setting of the searched architecture.** For the training setting on CIFAR-10, we follow the convention of (Lu et al., 2021b). As for the training setting on ImageNet, we train the searched architecture under two types of training settings to compare with other methods fairly. In the first setting, we train architecture for 350 epoch with the batch size of $512$. Besides, we use the SGD optimizer with a momentum of $0.9$, an initial learning rate of $0.2$, and the weight decay of $4 \times 10^{-5}$. In the second setting, the cosine learning rate strategy is used during the training on the basis of the first setting. In addition, we use some enhancement techniques to improve the classification accuracy on ImageNet. Specifically, label smoothing (Szegedy et al., 2016) and autoaugment (Cubuk et al., 2019) which are effective regularization methods are utilized in the training. Furthermore, the Squeeze-and-Excitation (SE) module (Hu et al., 2018) is attached after each cell.

## B    SUPPLEMENTARY EXPERIMENTS

In this part, we perform additional experiments on cell-based search space and block-based search space, and compare our method with other domain adaptation methods. The detail of the cell-based search space and the block-based search space used are shown in Fig. 6 and Fig. 7, respectively. Specifically, NAS-Bench-101, NAS-Bench-201, and DARTS belong to the cell-based search space, while NDS ResNet and NDS ResNeXt belong to the block-based search space.

Table 6: The description of three cell-based search spaces.

| Search space | Size | Operation Position | #Node | #Edge | #Operation type |
|---|---|---|---|---|---|
| NAS-Bench-101 | 423k | Node | 7 | 9 | 3 |
| NAS-Bench-201 | 15.6K | Edge | 4 | 6 | 5 |
| DARTS | $10^{18}$ | Edge | 6 | 8 | 7 |

Table 7: The description of two block-based search spaces.

| Search space | Size | Depth | Width | Ratio | Group number |
|---|---|---|---|---|---|
| NDS ResNet | 1260k | 9 | 12 | - | - |
| NDS ResNeXt | 11391k | 5 | 5 | 3 | 3 |

## B.1 CELL-BASED SEARCH SPACE

Table 8: The KTau value of USPP on NAS-Bench-201 (NAS-Bench-101→NAS-Bench-201).

| Methods | KTau |
|---|---|
| MMD (Gretton et al., 2006) | 0.406 |
| DANN (Ganin & Lempitsky, 2015) | 0.500 |
| GVB (Cui et al., 2020b) | 0.496 |
| HDA (Cui et al., 2020a) | 0.516 |
| **USPP (ours)** | **0.655** |

Table 9: The KTau value of USPP on NAS-Bench-101 (NAS-Bench-201→NAS-Bench-101).

| Methods | KTau |
|---|---|
| MMD (Gretton et al., 2006) | 0.306 |
| DANN (Ganin & Lempitsky, 2015) | 0.325 |
| GVB (Cui et al., 2020b) | 0.302 |
| HDA (Cui et al., 2020a) | 0.307 |
| **USPP (ours)** | **0.349** |

**Transferring from NAS-Bench-101 to NAS-Bench-201.** The comparisons of the KTau value on NAS-Bench-201 between USPP and other domain adaptation methods are shown in Table. 9. It can be seen that the USPP gets the highest KTau value among all peer competitors. This illustrates that the proposed method can effectively transfer knowledge from NAS-Bench-101 to NAS-Bench-201.

**Transferring from NAS-Bench-201 to NAS-Bench-101.** Table 8 shows the comparison results of USPP on NAS-Bench-101. It can be seen that USPP achieves the highest KTau value among all peer competitors, which verifies the effectiveness of USPP to transfer from NAS-Bench-201 to NAS-Bench-101.

**Transferring from NAS-Bench-201 to DARTS.** USPP is evaluated in NAS-Bench-301. The comparison results of transferring from NAS-Bench-201 to DARTS search space are shown in Table. 10. It can be observed that although USPP surpasses all competitors, the KTau value of USPP is low. We infer that this is because NAS-Bench-201 is too small compared to DARTS (15.6k vs $10^{18}$), and the domain discrepancy between this NAS-Bench-201 and DARTS is too large. The information learned from NAS-Bench-201 is not enough to accurately predict the architecture in DARTS.

Table 10: The KTau value of USPP on DARTS search space (NAS-Bench-201→DARTS).

| Methods | KTau |
|---|---|
| MMD (Gretton et al., 2006) | 0.160 |
| DANN (Ganin & Lempitsky, 2015) | 0.185 |
| GVB (Cui et al., 2020b) | 0.255 |
| HDA (Cui et al., 2020a) | 0.153 |
| **USPP (ours)** | **0.287** |

Table 13: The regression error and the domain classification accuracy.

| Loss and accuracy | Value |
|---|---|
| Regression error | 0.056 |
| Domain classification accuracy | 0.325 |

## B.2 BLOCK-BASED SEARCH SPACE

Table 11: The KTau value of USPP on NDS ResNet (NDS ResNeXt→NDS ResNet).

| Methods | KTau |
|---|---|
| MMD (Gretton et al., 2006) | 0.373 |
| DANN (Ganin & Lempitsky, 2015) | 0.334 |
| GVB (Cui et al., 2020b) | 0.427 |
| HDA (Cui et al., 2020a) | 0.239 |
| **USPP (ours)** | **0.442** |

Table 12: The KTau value of USPP on NDS ResNeXt (NDS ResNet→NDS ResNeXt).

| Methods | KTau |
|---|---|
| MMD (Gretton et al., 2006) | 0.519 |
| DANN (Ganin & Lempitsky, 2015) | 0.513 |
| GVB (Cui et al., 2020b) | 0.485 |
| HDA (Cui et al., 2020a) | 0.391 |
| **USPP (ours)** | **0.616** |

**Transferring from NDS ResNet to NDS ResNeXt.** We apply the 25k public annotated architectures in the NDS ResNeXt search space and the unlabeled architectures in the NDS ResNet search space to train USPP. Please note that we do not use the proposed operation embedding method, and replace the GCN (i.e., feature extractor) with the MLP. The comparison results of the KTau value are shown in Table. 11. The results show that the proposed USPP surpasses all competitors. This verifies the effectiveness of our approach in NDS ResNet which belongs to the block-based search space.

**Transferring from NDS ResNeXt to NDS ResNet.** Similar to the last experiment (i.e., NDS ResNeXt→NDS ResNet), We train USPP on 25k labeled architectures in the NDS ResNet search space and test USPP on the public annotated architectures in the ResNeXt search space. The comparison results are shown in Table 12. USPP achieves the highest KTau value. This illustrates that the proposed method can effectively learn the correlation between architecture and performance and transfer it to the NDS ResNeXt search space.

## C DETAILS OF THE TRAINING PROCESS

**Quantity of the domain-adaptation features.** We have performed experiments on NAS-Bench-201 → DARTS and NAS-Bench-101 → NAS-Bench-201, and computed the mean KL-Divergence and the mean cosine similarity. Specifically, the KL-Divergence is 0.0086, and the mean cosine similarity is 0.5018. These results show that the domain invariant features we learn are similar in different cases (i.e., NAS-Bench-201 → DARTS, NAS-Bench-101 → NAS-Bench-201), and further prove the quality of domain invariant features.

**The regression error and the domain classification accuracy.** The results are shown in Table. 13. The regression error is low, and illustrates that the regressor learns the correspondence between the architecture and performance in the source domain. Furthermore, the domain classification accuracy can verify that the discriminator cannot distinguish between domains.

