# OpenReview forum: "Unsupervised Performance Predictor for Architecture Search"
_ICLR.cc/2023/Conference — Submitted to ICLR 2023_

### Official Review · Reviewer_4ZdJ · 2022-10-21

**Confidence:** 4
**Correctness:** 4
**Technical Novelty And Significance:** 2
**Empirical Novelty And Significance:** 2
**Recommendation:** 5

**Clarity, Quality, Novelty And Reproducibility:**

This paper is written very clearly. But the novelty of this paper is limited. The idea is heavily borrowed from the paper "Gradually Vanishing Bridge for Adversarial Domain Adaptation" by Cui et al. That paper uses a bridge layer for both generator and discriminator and use the domain invariant features to compute the classification loss and adversarial loss.



**Strength And Weaknesses:**

**Strengths**:
1. The paper is able to learn a surrogate model on the target search space without any training data

**Questions**:
1. Could you evaluate the quality of the domain invariant features that are learnt? For example, use the source domain as nasbench-201 and target as Darts, source domain as nasbench-101 and target as nasbench-201 and compute the distance (KL-Divergence) or similarity (cosine similarity) between the learnt domain invariant features using source domain 1 and source domain 2.
2. Could you also include a table for the regression error and the domain classification  accuracy?
3. Is USPP DANN + bridge layer. If so, in the ablation study, when the bridge layer is removed, why is the kendall tau different from DANN? For table 5, are you using operation embeddings for all the baselines?
4. For table 5, rather than training the architectures for darts search space, why not use NDS-Darts [1] benchmark? Evaluate it on 100 architectures sampled from that search space.  Similarly, for table 2, if you could actually use NDS-Darts to find the best architecture and report the numbers for other algorithms also from the same benchmark, it would be a fair comparison.
5. Please include darts-pt [2] also in table 1 and 2.

[1] On Network Design Spaces for Visual Recognition, Radosavovic et al.
[2] Rethinking Architecture Selection in Differentiable NAS, Wang et al.

**Summary Of The Paper:**

This paper presents a surrogate model that predicts the accuracy of an architecture.  The surrogate model has access to the architectures and their corresponding accuracies from the source search space. It is tasked with predicting the accuracies of architectures that are sampled from the target search space. It uses adversarial training for unsupervised domain adaptation. A Graph convolution network is used to extract the latent features from the architectures and serves as a generator. The discriminator in turn tries to differentiate the domain of the architecture.  A fundament layer F models the entire feature space.  A bridge layer B captures the domain specific features. The difference between the $F(x_{i})$ and $B(x_{i})$ gives the domain invariant features. The domain invariant features are fed to a predictor layer which in turn predicts the accuracy.

The overall loss function is a combination of the domain loss from the discriminator, the bridge layer loss and the regression loss from the predict layer. The input to the surrogate model is an adjacency matrix. It learns an embedding for the operations of the architecture.

**Summary Of The Review:**

Domain adaptation to learn a surrogate model for NAS is very useful for practical purposes. But the paper is mainly adapted from "Gradually Vanishing Bridge for Adversarial Domain Adaptation". Using operations for embeddings has also been done in the past. So the novelty is limited.

---

> ### Author Response · Authors · 2022-11-17
> **Reply to Reviewer 4ZdJ (Part 1/2)**
>
> Dear reviewer, thank you sincerely for your time and efforts in reviewing our paper and constructive comments. Please kindly find our responses to your raised questions below. We hope our response can address your concerns.
> ***
> > Could you evaluate the quality of the domain invariant features that are learnt? For example, use the source domain as nasbench-201 and target as Darts, source domain as nasbench-101 and target as nasbench-201 and compute the distance (KL-Divergence) or similarity (cosine similarity) between the learnt domain invariant features using source domain 1 and source domain 2.
>
> Yes, we can. We have performed experiments on NAS-Bench-201 $\rightarrow$ DARTS and NAS-Bench-101 $\rightarrow$ NAS-Bench-201, and computed the mean KL-Divergence and the mean cosine similarity. Specifically, the KL-Divergence is 0.0086, and the mean cosine similarity is 0.5018. These results show that the domain invariant features we learn are similar in different cases (i.e., NAS-Bench-201 $\rightarrow$ DARTS, NAS-Bench-101 $\rightarrow$ NAS-Bench-201), and further prove the quality of domain invariant features.
> ***
> > Could you also include a table for the regression error and the domain classification accuracy?
>
> In response to the comments made by this reviewer, we have included a table for the regression error and domain classification accuracy. The regression error is low, and illustrates that the regressor learns the correspondence between the architecture and performance in the source domain. Furthermore, the domain classification accuracy can verify that the discriminator cannot distinguish between domains.
>
> |  Loss  | Value|
> |  ----    | ----  |
> |  Regression error   | 0.056 |
> | Domain classification accuracy     | 0.325 |
>
> In addition, we have also provided the figures of how the regression error, domain classification error, and bridge error vary as the improvement of iteration number in Fig. 2.
>
> ***
> > 1\) Is USPP DANN + bridge layer. If so, in the ablation study, when the bridge layer is removed, why is the kendall tau different from DANN? 2\) For table 5, are you using operation embeddings for all the baselines?
>
> 1) No, USPP=DANN+bridge layer+operation embedding. The kendall tau is different from DANN because the operation embedding still exists when the bridge layer is removed.
> 2) We do not use the designed operation embedding in the baseline.
>
> ***
> > 1\) For table 5, rather than training the architectures for darts search space, why not use NDS-Darts [1] benchmark? Evaluate it on 100 architectures sampled from that search space. 2\) Similarly, for table 2, if you could actually use NDS-Darts to find the best architecture and report the numbers for other algorithms also from the same benchmark, it would be a fair comparison.
>
> 1) We could use NDS-Darts benchmark for the comparisons. However, NDS-Darts has not been widely used in the community, and most states of the arts did not report their results on NDS-Darts. In this regard, using NDS-Darts may be not suitable. In practice, NAS-Bench-301 is similar to NDS-Darts and more popular among the community. As a result, we have chosen the NAS-Bench-301 benchmark to perform the experiments, and calculated the KTau value for comparisons. The results are shown below. As can be seen, USPP still achieves the highest KTau, which verifies the effectiveness of USPP.
>
>     |Domain adaptation method |KTau|
>     |----|----|
>     |MMD|0.3162|
>     |DANN|0.2424|
>     |GVB|0.2699|
>     |HDA|0.3173|
>     |**USPP (ours)**|**0.4113**|
>
> 2) We are sorry that NDS-Darts cannot be used to perform the experiments in table 2. This is because NDS-Darts only includes 5k annotated architectures while the DARTS search space contains $10^{18}$ architectures, which means the best architecture in the DARTS search space may not be in NDS-Darts. This fact is also applied to NAS-Bench-301 which has 11k annotated architectures, which is far less than that in the DARTS search space.
>
> [1] Radosavovic, Ilija, et al. "On network design spaces for visual recognition." Proceedings of the IEEE/CVF international conference on computer vision. 2019.
>
>
> ***
> > Please include darts-pt [2] also in table 1 and 2.
>
> Thanks for your suggestion. We have included DARTS-PT in Tables 1 and 2 and shown below. As can be seen, the proposed USPP method has higher classification accuracy and fewer GPU days than DARTS-PT.
>
> * Comparison in DARTS search space on CIFAR-10.
>
>     |Method|Accuracy (%)|#Param. (M)|Cost (GPU day)|
>     |----|----|----|----|
>     |DARTS-PT|97.54|3.9|0.29|
>     |**USPP(ours)**|**97.86**|**3.3**|**0.02**|
>
>
> * Comparison in DARTS search space on ImageNet.
>
>     |Method|Top-1 (%)|Top-5 (%)|#Param. (M)|Cost (GPU day)|
>     |----|----|----|----|----|
>     |DARTS-PT|74.5|92.0|4.6|0.29|
>     |**USPP(ours)**|**75.9**|**92.9**|5.0|**0.02**|

---

> > ### Author Response · Authors · 2022-11-17
> > **Reply to Reviewer 4ZdJ (Part 2/2)**
> >
> > > Domain adaptation to learn a surrogate model for NAS is very useful for practical purposes. But the paper is mainly adapted from "Gradually Vanishing Bridge for Adversarial Domain Adaptation" (GVB). Using operations for embeddings has also been done in the past. So the novelty is limited.
> >
> > Thanks for pointing out this concern. Although the bridge layer and the operation embedding exist in the previous paper, both components are not simply combined to form the proposed USPP method. First, although we use the bridge layer, the basic framework of the domain adaptation and the position of the bridge layer is completely different from GVB. Furthermore, the reason why we use the bridge layer for the prediction layer is that the discrepancy between different domains for architecture data is direct and significant and this makes it easier for the bridge layer to model and reduce the discrepancy between different domains. Second, the operation embedding is used as an auxiliary component. It can learn the intrinsic relationship between operations in the source domain and apply it in the target domain, thus transferring more knowledge to the target domain. In previous papers, the operation embedding is not designed for domain adaptation. In fact, as far as we know, the adversarial domain adaptation technique (including the bridge layer) and the operation embedding are both used in the performance predictor for the first time.
> >
> > Last but not least, the biggest innovation of our paper is to ease the huge computational cost of neural architecture search via unsupervised domain adaptation, as recognized by this reviewer "Domain adaptation to learn a surrogate model for NAS is very useful for practical purposes". We only cost 0.02 GPU day to search for a promising architecture. We also have made a preliminary exploration for transferring between various search spaces such as NAS-Bench-101, NAS-Bench-201, DARTS search space, NDS ResNet, and NDS ResNeXt.

---

### Official Review · Reviewer_YaG5 · 2022-10-23

**Confidence:** 4
**Clarity, Quality, Novelty And Reproducibility:** The paper is well-written and easy to…
**Correctness:** 3
**Technical Novelty And Significance:** 3
**Empirical Novelty And Significance:** 3
**Recommendation:** 5

**Strength And Weaknesses:**

Strengths:
1. This paper develops a new NAS method that trains a NAS model and a regressor model in an adversarial scheme.
2. The proposed progressive domain-invariant feature extraction method is interesting.


Weaknesses:

1. There is a typo in the abstract. The accuracy on ImageNet should be 76.50% instead of 96.50%.

2. As highlighted in the paper, one of the contributions is the learnable operation embedding. However, it is not novel because almost all Reinforcement Learning (RL) based NAS methods exploit a learnable embedding for each operation, such as ENAS. Please further clarify it if there are some other differences from what is used in ENAS.

3. The motivation for reducing the divergence between source and target search space seems questionable. For example, in Figure 1, the architectures in source and target domains may share exactly the same or similar architecture. In other words, the discriminator would definitely fail in this case since it cannot distinguish between two identical/similar architectures. Instead, what should be bridged/distinguished between two spaces is the accuracy y instead of the architecture x.

4. The application scenario of the proposed method seems very limited. According to the paper, this method can be only used on the search space of DARTS. Given another search space, e.g., MobileNet based space used in OFA, is the proposed method still applicable in this case?



**Summary Of The Paper:**

This paper proposes an unsupervised performance predictor called USPP to reduce the training/search cost of NAS. To bridge the source and target search spaces, the authors develop a progressive domain-invariant feature extraction method to obtain domain-invariant features of architectures. Moreover, this paper provides sufficient ablation studies for each module/element of the proposed method. Nevertheless, both the novelty of the learning operation embedding and the application scenario of the proposed method are very limited.

**Summary Of The Review:**

The application scenario of the proposed method seems very limited.

---

> ### Author Response · Authors · 2022-11-17
> **Reply to Reviewer YaG5 (Part 1/2)**
>
> Dear reviewer, thank you sincerely for your time and efforts in reviewing our paper and constructive comments. Please kindly find our responses to your raised questions below. We hope our response can address your concerns.
> ***
> > There is a typo in the abstract. The accuracy on ImageNet should be 76.50% instead of 96.50%.
>
> Thanks for your reminder, and we have revised it in the modified version.
> ***
> > As highlighted in the paper, one of the contributions is the learnable operation embedding. However, it is not novel because almost all Reinforcement Learning (RL) based NAS methods exploit a learnable embedding for each operation, such as ENAS. Please further clarify it if there are some other differences from what is used in ENAS.
>
> Thanks for pointing out this issue.
> In fact, the principle of learnable operation embedding in ENAS is different from that of our operation embedding.
>
> In ENAS, the controller receives an empty embedding as input, and the decision (i.e., operation) in the previous step is fed as input embedding into the next step. Just as says in ENAS, "We utilize three regularization techniques, ... tying word embeddings [1].", the learnable embedding for operation (i.e., word embeddings) is a common regularization technique for RNN (the controller of ENAS) but not designed for RL-based NAS. Researchers generally think that the learnable embedding in RL-based NAS can indicate the effect of a local operation on the whole network [2]. However, there is no RL-based NAS paper to study why it works.
>
> In USPP, the operation embedding is specially designed as an auxiliary component for our domain adaptation method. This is because we observe that the learned operation embedding can implicitly represent the internal relationship between operations, and the application of the operation embedding learned from the source domain will help improve the prediction accuracy in the target domain.
> The 3D visual results of the resulting embedding in Figure. 2 verify that similar operations have smaller distances, and the ablation study illustrates that the operation embedding can improve the prediction performance of USPP.
>
> [1] Inan, Hakan, Khashayar Khosravi, and Richard Socher. "Tying word vectors and word classifiers: A loss framework for language modeling." arXiv preprint arXiv:1611.01462 (2016).
>
>
> [2]Chen, Yukang, et al. "Renas: Reinforced evolutionary neural architecture search." Proceedings of the IEEE/CVF Conference on Computer Vision and Pattern Recognition. 2019.
>
> ***
> > The motivation for reducing the divergence between source and target search space seems questionable. 1) For example, in Figure 1, the architectures in source and target domains may share the same or similar architecture. In other words, the discriminator would fail in this case since it cannot distinguish between two identical/similar architectures. 2) Instead, what should be bridged/distinguished between two spaces is the accuracy $y$ instead of the architecture $x$.
>
> Thanks for pointing out this concern. We divide the question into two parts and answer them separately.
> 1) The discriminator is only an auxiliary component to help the feature extractor extract the domain-invariant features. If the architectures in the source and target domains share the same or similar architecture, the discrepancy between the two domains is smaller and it is easier to extract domain-invariant features. The feature extractor can easily fool the discriminator and the discriminator naturally fails to distinguish between similar architectures. This is exactly what we want to see because the goal of adversarial training is to extract domain-invariant features.
>
> 2) Distinguishing between two spaces by accuracy $y$ cannot achieve the goal of extracting domain-invariant features. This is because the domain label (e.g., 0 for the source domain, and 1 for the target domain) for the extracted feature is definite while the accuracy for the target domain is unknown. In addition, since the accuracy in the source domain and target domain is standardized, we cannot judge whether the architecture belongs to the source domain or target domain only by the accuracy value. As a result, the accuracy $y$ cannot be used instead of the architecture $x$.

---

> > ### Author Response · Authors · 2022-11-17
> > **Reply to Reviewer YaG5 (Part 2/2)**
> >
> > > The application scenario of the proposed method seems very limited. According to the paper, this method can be only used on the search space of DARTS. Given another search space, e.g., MobileNet based space used in OFA, is the proposed method still applicable in this case?
> >
> > Yes, it is still applicable in this case. Because there is no benchmark on the MobileNet search spaces, we have added the experiments on two MobileNet-like spaces, i.e., NDS ResNet and NDS ResNeXt search spaces [3]. Similar to MobileNet search space, the NDS ResNet and NDS ResNeXt search spaces have many choices of block decisions (e.g., the number of channels) for each block in the architectures. As a result, we regarded NDS ResNet and NDS ResNeXt search spaces as MobileNet-like spaces. The experimental results on NDS ResNet $\rightarrow$ NDS ResNeXt and NDS ResNeXt $\rightarrow$ NDS ResNet are shown in the following tables. It can be seen that the proposed method surpasses all other domain adaptation methods, which can verify the broad application scenario of the proposed method.
> >
> > * The KTau value of USPP on NDS ResNeXt $\rightarrow$ NDS ResNet and NDS ResNet $\rightarrow$ NDS ResNeXt.
> >
> >     |Methods|ResNeXt $\rightarrow$ ResNet|ResNet $\rightarrow$ ResNeXt|
> >     |----| ---- | ---- |
> >     | MMD| 0.373|0.519 |
> >     |DANN| 0.334|0.513 |
> >     | GVB| 0.427|0.485 |
> >     | HDA| 0.239|0.391 |
> >     |**USPP (ours)**|**0.442**|**0.616**|
> >
> >
> > [3] Radosavovic, Ilija, et al. "On network design spaces for visual recognition." Proceedings of the IEEE/CVF international conference on computer vision. 2019.
> >
> > ***

---

### Official Review · Reviewer_8UES · 2022-10-27

**Confidence:** 4
**Clarity, Quality, Novelty And Reproducibility:** 1. The paper is well-written and easy…
**Correctness:** 3
**Technical Novelty And Significance:** 3
**Empirical Novelty And Significance:** 3
**Recommendation:** 5

**Strength And Weaknesses:**

Pros:
1. The paper is well written and easy to follow.
2. The proposed method is technically sound.
3. The idea of using UDA and adversarial training to address the domain gap issue in NAS is smart.

Cons:
1. The name of the paper is a little bit confusing at the first glance. I thought it is a method related to unsupervised learning, but turns out to be related to Unsupervised Domain Adaptation.
2. Missing recent baselines in the result table[1][2].
3. The author uses GPU day as a metric to evaluate searching speed. My question is that are the compared methods using the same GPU? If not, then they are not comparable.
4. The author only report one domain transfer result (NAS101 --> DARTS). I would expect to see more domain transfer case, like NAS101 --> NAS102,NAS102 -->DARTS etc.

[1] Mellor, J., Turner, J., Storkey, A., & Crowley, E. J. (2021, July). Neural architecture search without training. In International Conference on Machine Learning (pp. 7588-7598). PMLR.
[2] hen, W., Gong, X., & Wang, Z. (2021). Neural architecture search on imagenet in four gpu hours: A theoretically inspired perspective. arXiv preprint arXiv:2102.11535.

**Summary Of The Paper:**

The paper proposed an unsupervised (UDA) performance predictor called USPP, which aims to mitigate the architecture performance prediction gap between source and target domain.

**Summary Of The Review:**

The overall quality of this paper is good. The proposed method has promising results on the setting of transferring NAS101 to DARTS search space. I would expect the author to conduct more transfer settings to validate the effectiveness of the proposed method. I would like to recommend this paper to score 6 temporarily.

---

> ### Author Response · Authors · 2022-11-17
> **Reply to Reviewer 8UES (Part 1/2)**
>
> Dear reviewer, thank you sincerely for your time and efforts in reviewing our paper and constructive comments. Please kindly find our responses to your raised questions below. We hope our response can address your concerns.
> ***
> > The name of the paper is a little bit confusing at the first glance. I thought it is a method related to unsupervised learning, but turns out to be related to Unsupervised Domain Adaptation.
>
> Thanks for your reminder. We have revised the title to "Performance Prediction via Unsupervised Domain Adaptation for Architecture Search".
> ***
> > Missing recent baselines in the result table[1][2].
>
> We have added both baselines, i.e., TENAS (i.e., reference [2]) and NASWOT (i.e., reference [1]), for the comparisons. Specifically, TENAS provided the results on both CIFAR-10 and ImageNet, so we directly use the results for the comparison. However, NASWOT did not provide the results on both datasets, and we got the results on CIFAR-10 from another paper [3]. Unfortunately, we currently do not have enough time to perform the experiment on ImageNet during this tight rebuttal period, and the experiment is still in process.
>
> The comparisons have been provided as shown in the following tables. Clearly can be seen, our proposed method also achieved promising results against both baselines in terms of accuracy, params, and GPU day. We will include the result of NASWOT on ImageNet in the camera-ready version if this paper has a chance to be accepted finally.
>
> * Comparison results on CIFAR-10.
>
>     |  Method  | Accuracy (%)| #Params. (M) | Cost (GPU day)|
>     |  ----    | ----  | ----|----|
>     |  TENAS   | 97.37 |3.8  |0.05|
>     |  NASWOT  | 95.73 |4.96 |-   |
>     | **USPP (ours)**     | **97.86** |**3.3**  |**0.02**|
>
> * Comparison results on ImageNet.
>
>     |  Method  | Top-1 (%)| Top-5 (%) | #Params.(M)| Cost (GPU day)|
>     |  ----    | ----  | ----|----|----|
>     |  TENAS   | 75.5 |92.5  |5.4 |0.17|
>     | **USPP (ours)**     | **75.9** |**93.0**  |**5.0** |**0.02**|
>
>
> [1] Mellor, J., Turner, J., Storkey, A., & Crowley, E. J. (2021, July). Neural architecture search without training. In International Conference on Machine Learning (pp. 7588-7598). PMLR.
>
> [2] hen, W., Gong, X., & Wang, Z. (2021). Neural architecture search on imagenet in four gpu hours: A theoretically inspired perspective. arXiv preprint arXiv:2102.11535.
>
> [3] Ning, Xuefei, et al. "Evaluating efficient performance estimators of neural architectures." Advances in Neural Information Processing Systems 34 (2021): 12265-12277.
> ***
> > The author uses GPU day as a metric to evaluate searching speed. My question is that are the compared methods using the same GPU? If not, then they are not comparable.
>
> The compared methods do not use the same GPU. We recognize that these comparisons may be not fair, however, this is a convention among the NAS community as can be seen from various related papers published. This is because reperforming these algorithms on the same GPU model is impractical, since each search method often used hundreds of GPUs.
>
> In this paper, we used NVIDIA RTX 2080Ti GPUs for conducting the experiments. **To further clarify the efficiency of the proposed method, we re-conducted the experiments with the 1080Ti GPU which has poorer performance.** Specifically, the NVIDIA 1080Ti has the poorest performance except for P100 and TITAN Xp. The comparison results are shown in the following table. We found that our proposed method only consumed **0.025 GPU days on 1080Ti**, and still has the lowest search cost. Furthermore, our GPU day is 5,5 and 2 orders of magnitude smaller than NASNet-A (P100), AmoebaNet-A (P100) and SNAS (TITAN Xp), respectively. The results illustrate that our method can ease the huge computational cost of NAS.
>
>
> |Method	|GPU type| GPU day (CIFAR-10)|GPU day (ImageNet)|
> | ---- | ----| ---- | ---- |
> |NASNet-A	|NVIDIA P100| 1,800| 2,000|
> |AmoebaNet-A	|NVIDIA P100| 3,150|3,150|
> |ENAS	|NVIDIA GTX 1080Ti| 0.5 |-|
> |DARTS	|NVIDIA GTX 1080Ti| 4| 4|
> |SNAS	|NVIDIA TITAN Xp| 1.5 | 1.5|
> |PC-DARTS	|NVIDIA Tesla V100| 0.1 |0.1|
> |P-DARTS	|NVIDIA Tesla V100| 0.3 |-|
> |PNAS	|NVIDIA P100| - |-|
> |DropNAS	|NVIDIA Tesla V100| - |0.6|
> |DARTS-PT	|NVIDIA GTX 1080Ti| 0.29|0.29|
> |NAONet	|NVIDIA Tesla V100 | 200|200|
> |BONAS	|NVIDIA Tesla V100 | 2.5|-|
> |CTNAS	|Not report       | 0.3 |-|
> |TNASP	|NVIDIA Tesla V100| 0.3 |-|
> |TENAS	|NVIDIA GTX 1080Ti| 0.05 |0.17|
> |**USPP(ours)**|NVIDIA RTX 2080Ti|**0.02**|**0.02**|
> |**USPP(ours)**|NVIDIA GTX 1080Ti|0.025|0.025|
> ***

---

> > ### Author Response · Authors · 2022-11-17
> > **Reply to Reviewer 8UES (Part 2/2)**
> >
> > > The author only reports one domain transfer result (NAS101 --> DARTS). I would expect to see more domain transfer case, like NAS101 --> NAS102, NAS102 -->DARTS etc.
> >
> > We concur with the suggestion given by the reviewer, and have performed the experiments of **NAS-Bench-101 $\rightarrow$ NAS-Bench-201, NAS-Bench-201 $\rightarrow$ NAS-Bench-101, NAS-Bench-201 $\rightarrow$ DARTS, NDS ResNet [4]$\rightarrow$ NDS ResNeXt [4]** (NDS ResNet and NDS ResNeXt are both block-based search space), and **NDS ResNeXt $\rightarrow$ NDS ResNet**. We compared the KTau value of USPP with other domain adaptation methods, and the results are shown below.
> >
> > The experimental results show that the proposed USPP method achieved the highest KTau among all peer competitors in NAS-Bench-101 $\rightarrow$ NAS-Bench-201, NAS-Bench-201 $\rightarrow$ NAS-Bench-101 and NDS ResNeXt $\rightarrow$ NDS ResNet, and NDS ResNet $\rightarrow$ NDS ResNeXt, and verify the superiority of USPP.
> >
> > * The KTau value of USPP.
> >
> >     |Methods|	101 $\rightarrow$ 201| 201 $\rightarrow$ 101| 201 $\rightarrow$ DARTS | ResNeXt $\rightarrow$ ResNet|ResNet $\rightarrow$ ResNeXt|
> >     | ----| ----| ----  | ----  |----  | ---- |
> >     |MMD  |0.406| 0.306 | 0.160 | 0.373| 0.519|
> >     |DANN |0.500| 0.325 | 0.185 | 0.334| 0.513|
> >     |GVB  |0.496| 0.302 | 0.255 | 0.427| 0.485|
> >     |HDA  |0.516| 0.307 | 0.153 | 0.239| 0.391|
> >     |**USPP (ours)**|**0.655**|**0.349**|**0.287**|**0.442**|**0.616**|
> >
> >
> > [4] Radosavovic, Ilija, et al. "On network design spaces for visual recognition." Proceedings of the IEEE/CVF international conference on computer vision. 2019.

---

### Author Response · Authors · 2022-11-17
**General Reply to Reviewers**

Dear reviewers,

We would like to thank you for taking the time and effort to review our manuscript. We have updated our paper with a revision. Please note extensive changes made in the modified version are shown in blue. In the modified version, we improve our manuscript in four aspects:

* **The title has been revised as "Performance Prediction via Unsupervised Domain Adaptation for Architecture Search"** to avoid confusing readers, as suggested by reviewer 8UES.

* **The experiments on NAS-Bench-101 $\rightarrow$ NAS-Bench-201, NAS-Bench-201 $\rightarrow$ NAS-Bench-101, and NAS-Bench-201 $\rightarrow$ DARTS search space** has been provided, as suggested by reviewer 8UES.

* **The experiments on NDS ResNet $\rightarrow$ NDS ResNeXt, NDS ResNeXt $\rightarrow$ NDS ResNet** have been provided to verify the broad application of our method, as suggested by reviewer YaG5.

* **More details about the training process of USPP** are provided, as suggested by reviewer 4ZdJ.

If you have new suggestions, please let us know.

Thanks,

Authors.

---

### Decision · Program_Chairs · 2023-01-20

**Decision:**

Reject

**Justification For Why Not Higher Score:**

Although the direction studied in the paper is interesting, all the reviewers think the current experiments are not sufficient to justify the use cases of the proposed method.

**Justification For Why Not Lower Score:**

N/A

**Metareview: Summary, Strengths And Weaknesses:**

The paper proposes an unsupervised performance predictor for NAS which uses a progressive domain-invariant feature extraction method to obtain domain-invariant features of architectures. Although the topic is interesting and can potentially reduce the NAS search cost, all the reviewers agree that the current experiments are not sufficient to show that the proposed method works in general cases. To justify the use cases of the proposed method, we suggest the authors conduct experiments on other sources and target domains, beyond transferring from NAS-bench to DARTS space.

**Summary Of Ac-Reviewer Meeting:**

The paper originally got 6, 5, 5, so we were planning to have a reviewer meeting for this paper. However, during our email communications, one reviewer decided to lower the score to 5, thus we reached a consensus to reject this paper.